# Hybrid Boron-Carbon Chemistry

**DOI:** 10.3390/molecules25215026

**Published:** 2020-10-29

**Authors:** Josep M. Oliva-Enrich, Ibon Alkorta, José Elguero

**Affiliations:** 1Instituto de Química-Física “Rocasolano” (CSIC), Serrano 119, E-28006 Madrid, Spain; 2Instituto de Quimica Médica (CSIC), Juan de la Cierva, 3, E-28006 Madrid, Spain; ibon@iqm.csic.es (I.A.); iqmbe17@iqm.csic.es (J.E.)

**Keywords:** boron, conjugated hydrocarbon, isoelectronic molecule, electronic structure, quantum chemistry, singlet-triplet gap

## Abstract

The recently proved one-to-one structural equivalence between a conjugated hydrocarbon C*_n_*H*_m_* and the corresponding borane B*_n_*H*_m_*_+*n*_ is applied here to hybrid systems, where each C=C double bond in the hydrocarbon is consecutively substituted by planar B(H_2_)B moieties from diborane(6). Quantum chemical computations with the B3LYP/*cc*-pVTZ method show that the structural equivalences are maintained along the substitutions, even for non-planar systems. We use as benchmark aromatic and antiaromatic (poly)cyclic conjugated hydrocarbons: cyclobutadiene, benzene, cyclooctatetraene, pentalene, benzocyclobutadiene, naphthalene and azulene. The transformation of these conjugated hydrocarbons to the corresponding boranes is analyzed from the viewpoint of geometry and electronic structure.

## 1. Introduction

Planar conjugated hydrocarbons played a key role in the early days of quantum mechanics, when computers were not available, and physical models were needed in order to understand the electronic structure of molecules [1], the attractive nature of the chemical bond [2], the nature of ground and excited states in atoms and molecules [3,4,5], and mechanistical studies of chemical reactions [6].

On the other hand, a relevant quantity in chemistry is the energy gap between the lowest-lying singlet and triplet electronic states in a molecule, directly related to the reactivity of the system, and useful, e.g., for the design of photochemical molecular devices [7]. For instance, methylene CH_2_ is a very reactive species with a triplet ground state and a singlet-triplet experimental energy gap of 38 kJ·mol^−1^ [8]. On the other hand, water H_2_O has a singlet ground state with an experimental singlet-triplet energy gap of 675 kJ·mol^−1^ [9]. Hence, the larger the singlet-triplet energy gap the more stable a molecule.

The chemistry of boron compounds has evolved along the second half of the XXth century and beginning of the XXIst century in a generalised dual fashion since the original works of Stock [10] on B*_n_*H*_m_* boranes syntheses in the beginning of the XXth century: (i) organoboron [11] and metal-boron [12] chemistry and (ii) the chemistry of 3D polyhedral (metalla)heteroborane structures [13,14,15,16,17]. Given the huge and magnificent efforts done towards the synthesis of new boron derivatives in the last decades, a complete literature citation here is impractical.

The recent experimental isolation of borophane layers [18], a 2D (BH)_1_ system structurally equivalent to graphene and the characterisation of chemical structures where one C=C double bond is substituted by one B(H_2_)B moiety [19,20,21,22,23] calls for the possibility of creating a new field of research within boron chemistry, namely, the synthesis of finite planar neutral borane molecules. We have recently proved that there is a one-to-one structural correspondence between any planar conjugated hydrocarbon C*_n_*H*_m_* and the planar borane B*_n_*H_(*m*+*n*)_, indeed with the same number of electrons and *n* more hydrogen atoms in the latter [24]. This transformation can be carried out by substituting all C=C double bonds by a perpendicular planar B(H_2_)B moiety, which is the central rhombus of diborane(6). Up to this date we have not found any exception so far to this transformation and is extended here to non-planar systems.

The problems that we would like to tackle in this work, dedicated to Professor Todd B. Marder on the occasion of his 65th birthday, are related to the stability of hybrid boron-carbon isoelectronic chemical structures built by consecutive substitution of C=C by B(H_2_)B moieties in (poly)cyclic conjugated hydrocarbons. Particularly, we have chosen examples with 4*n* π electrons: cyclobutadiene, cyclooctatetraene, pentalene and benzocyclobutadiene, and examples with (4*n* + 2) π electrons: benzene, naphthalene and azulene. The question we would like to answer here is: Given a (poly)cyclic conjugated hydrocarbon C*_n_*H*_m_* with an even number of carbon atoms or π electrons, how similar or different are the hybrid systems C_(*n*−2*k*)_B_(2*k*)_H_(*m*+2*k*)_, with *k* = {0, 1, 2, …, *n*/2}, to the original hydrocarbon from the structural and electronic structure point of view? In Scheme 1 we gather the conjugated hydrocarbons considered in this work.

## 2. Results

Table 1, Table 2, Table 3, Table 4 and Table 5 gather the formula, structure, electronic energy, vertical singlet-triplet energy gaps and point-group symmetry (PGS) for all the systems included in this work, derived from a conjugated hydrocarbon C*_n_*H*_m_* with an even number of carbon atoms or π electrons and the structures derived from consecutive *k* { C=C ↔ B(H_2_)B } substitutions, with 1 ≤ *k* ≤ *n*/2. The triplet energy is computed with optimized geometry of the singlet ground state due to considerable geometrical changes in triplet optimizations, and therefore singlet-triplet energies are vertical. From cyclooctatetraene onwards, all C_(*n*-2*k*)_B_(2*k*)_H_(*m*+2*k*)_ structures have more than one isomer, for 1 ≤ *k* < *n*/2. Hence, one might inquire the extent of change in energy differences and singlet-triplet energies in different isomers of a given structure. When isomers arise under *k* {C=C ↔ B(H_2_)B} substitutions in a given structure, these are labelled as {(**I**), (**II**), …} and ordered in increasing energy, as displayed in Figure 1, Figure 2, Figure 3, Figure 4 and Figure 5. All structures and isomers correspond to planar structures, except for cyclooctatetraene C_8_H_8_ (*k* = 0) and structures C_(8−2*k*)_B_(2*k*)_H_(8+2*k*)_ (*k* = 1–4). All structures and isomers presented in this work, as computed with the B3LYP/cc-pVTZ model, correspond to energy minima. B3LYP/cc-pVTZ optimised geometries, in cartesian coordinates (Å), of the systems included in the work are presented in Appendix A.

At this point we should emphasize the geometrical changes that a C*_n_*H*_m_* conjugated hydrocarbon undergoes through *k* {C=C ↔ B(H_2_)B} substitutions, with *k* = {0, 1, 2, …, *n*/2}, in a given hybrid boron-carbon C_(*m*−2*k*)_B_(2*k*)_H_(*n*+2*k*)_ structure.

In Table 1 and Figure 1, we gather the structures, energies and singlet-triplet gaps for cyclobutadiene, benzene and cyclooctatetraene and the corresponding C_(*n*-2*k*)_B_(2*k*)_H_(*m*+2*k*)_ structures. We start off with the series for *n* = 4 in cyclobutadiene: C_(4−2*k*)_B_(2*k*)_H_(4+2*k*)_, with *k* = {0, 1, 2}. Given the antiaromatic nature of cyclobutadiene, this molecule is very reactive with tendency to dimerize and can be observed by matrix isolation below 35 K [26]. Substitution of one {C=C} moiety by one {B(H_2_)B} moiety in C_4_H_4_ (*k* = 1) leads to a C_2_B_2_H_6_ cyclic structure, with a singlet-triplet energy gap 140 kJ·mol^−1^ larger than in C_4_H_4_. A second {C=C ↔ B(H_2_)B } substitution in C_4_H_4_ (*k* = 2) leads to cyclic tetraborane(8), with a even larger singlet-triplet energy gap, 170 kJ·mol^−1^ higher than in C_4_H_4_.

The next system to be analyzed is aromatic benzene (*n* = 6) and the boron-carbon hybrids: C_(6−2*k*)_B_(2*k*)_H_(6+2*k*)_, with *k* = {0, 1, 2, 3}, as gathered in Table 1 and Figure 1. Benzene is a colorless liquid, with a characteristic odor, volatile, very flammable and carcinogenic. This molecule is the paradigm of Hückel theory [1] and the concept of aromaticity itself, with a very interesting debate reaching our days on the role of its correlation energy and the π electron spin-pairing [27]. As opposed to the previous example, the singlet-triplet energy gaps for benzene and cyclic hexaborane(12) B_6_H_12_ are quite similar, 8 kJ·mol^−1^ larger in benzene, due to the aromatic nature of the latter. The first {C=C ↔ B(H_2_)B} substitution in benzene leading to C_4_B_2_H_8_ (*k* = 1) decreases the singlet-triplet energy gap by more than 100 kJ·mol^−1^—due to aromaticity loss—but with further {C=C ↔ B(H_2_)B} substitutions the singlet-triplet energy gaps increase by 35 kJ·mol^−1^ and 60 kJ·mol^−1^ for C_2_B_4_H_10_ (*k* = 2) and B_6_H_12_ (*k* = 3) respectively. This is a clear indication that breaking the aromaticity of benzene leads, to a first instance, to a less stable structure. However, this stability is increased by along the series for *k* = {2, 3}, and three {C=C ↔ B(H_2_)B} substitutions leads to a structure—D_3h_ cyclic hexaborane(12) B_6_H_12_—with a singlet-triplet energy gap which is only 8 kJ·mol^−1^ lower as compared to benzene. This result is striking and remains at the very origin of the recent proposal we have put forward [24]: To every (poly)cyclic planar conjugated hydrocarbon C*_n_*H*_m_* there corresponds a boron equivalent structure B*_n_*H*_m_*_+*n*_ which is also an energy minimum.

Inclusion of empirical dispersion corrections with the B97D functional [28] shows a systematic decrease of the singlet-triplet energy gaps as compared to the B3LYP results, following a similar tendency, except for the complete borane structures B*_n_*H*_m_*_+*n*_ with considerably lowering of 47 kJ·mol^−1^ for cyclic hexaborane(12), as compared to the B3LYP results. For the cyclooctatetraene series the lowering is also noteworthy when including dispersion corrections in the functional; the structure of the stationary points—energy minima—is maintained with both functionals. As described below in the Discussion section, the bond distances are slightly elongated when dispersion corrections are included, which is in agreement with lower singlet-triplet energy gaps.

In order to highlight the electronic structure differences in benzene and its borane hybrids we have plotted in Table 1 the molecular electrostatic potential (MEP) for the benzene series C_(6−2*k*)_B_(2*k*)_H_(6+2*k*)_, with *k* = {0, 1, 2, 3}: The blue and red colors in the MEP indicate negative and positive charge attraction areas: clearly, the benzene π-electron cloud is a positive-charge attraction area [29,30] and the borane substituted areas are negative-charge attraction areas, with a complete blue area above the boron-atoms plane in cyclic hexaborane(12) B_6_H_12_. Just below the MEP in Table 1 we show the anisotropy of the induced current density (ACID) [31] for the same systems, with a clear blocking of the current in the region where a {C=C ↔ B(H_2_)B} substitution is carried out; therefore π electron delocalization lowers considerably as *k* increases.

We turn now to an antiaromatic system with *n* = 8 π electrons, cyclooctatetraene and the structural isoelectronic series summarized in Table 1 and Figure 1: C_(8−2*k*)_B_(2*k*)_H_(8+2*k*)_, with *k* = {0, 1, 2, 3, 4}. Cyclooctetraene is a colorless to light yellow flammable liquid at room temperature and adopts a non-planar *D_2d_* structure in the ground state [32,33]. We should emphasize that cyclooctatetraene C_8_H_8_ (*k* = 0) and the corresponding borane equivalent B_8_H_16_ (*k* = 4) are transition state structures in the planar D_4h_ conformation with energy barriers of 44.0 kJ·mol^−1^ [34] and 55.8 kJ·mol^−1^ respectively (B3LYP/cc-pVTZ computations), and thus the barrier for the *D_2d_* → *D_4h_* process is larger for the borane compound. Both energy minima correspond to D_2d_ structures, and this is the only series of non-planar hydrocarbon and hybrid boron-carbon structures included in this work. The main purpose of including cyclooctatetraene is due to the non-planarity of the system, which was not considered before [21].

Consecutive {C=C ↔ B(H_2_)B} substitutions in cyclooctatetraene C_8_H_8_ (*k* = 0) leads to C_(6−2*k*)_B_(2*k*)_H_(6+2*k*)_ structures with a larger singlet-triplet energy gap as *k* increases from zero to 4, with a final gap which is 170 kJ·mol^−1^ larger as compared to the original hydrocarbon. Cyclooctatetraene is the first instance where we have two different positional isomers in C_4_B_4_H_12_ (*k* = 2): Isomers **I** and **II**, with lower energy and larger singlet-triplet gap for the more symmetric structure **I** with C_2v_ symmetry. We define *N*_iso_ as the number of isomers for a given *k* in C_(*n*__−__2*k*)_B_(2*k*)_H_(*m*+2*k*)_. For instance, for azulene and *k* = 2—see Table 5 below, we have a set of ten C_6_B_4_H_12_ isomers: {**I**, **II**, **III**, …, *N*_iso_}, with *N_iso_* =10 in Roman numerals, i.e., *N*_iso_ = **X**.

We consider next the pentalene molecule C_8_H_6_, a two-pentagon one-side fused conjugated system, also with *n* = 8 π electrons, and therefore antiaromatic. Pentalene dimerizes at 173 K [35], and its derivative, 1,3,5-tri-tert-butylpentalene, has been synthesized [36], as a stabilized planar 8π electron system. In Table 2 and Figure 2 we gather the results for the pentalene series: C_(8−2*k*)_B_(2*k*)_H_(6+2*k*)_, with *k* = {0, 1, 2, 3, 4}. Again, due to the antiaromatic nature of the hydrocarbon, the singlet-triplet energy gap in the equivalent boron structure (*k* = 4) is 157 kJ·mol^−1^ higher in energy, as in the previous antiaromatic cases for cyclobutadiene and cyclooctatetraene, and the energy gap in pentalene is the lowest along the whole series of structures and isomers derived from C_(8−2*k*)_B_(2*k*)_H_(6+2*k*)_ and 1 < *k* ≤ 4, as the encircled value shown at the top of Figure 2.

The singlet-triplet energy gaps increase along the series for 0 ≤ *k* ≤ 3; the equivalent boron structure B_8_H_14_ (*k* = 4) has a lower gap as compared to the previous (*k* = 3) molecule C_2_B_6_H_12_. There are striking differences in the energy gaps as function of *k*, the number of {C=C ↔ B(H_2_)B} substitutions, and also within the set of isomers of a given *k*, as shown in Figure 2. The largest and lowest singlet-triplet energy gaps in the whole series are shown in Figure 2 as encircled numbers and with the corresponding structure. All structures and isomers for 1 ≤ *k* ≤ 4 have a larger singlet-triplet energy gap as compared to original pentalene (*k* = 0), a clear indication on how unstable this conjugated hydrocarbon is as compared to the boron-carbon hybrid series.

A shown in Table 2 and Figure 2 the isomer energies for a given structure (*k*) follow the same order as the singlet-triplet gaps, except isomer **IV** for *k* = 2 with an increase of 100 kJ·mol^−1^. It is striking that isomers with very close electronic energies might have such different singlet-triplet gaps. The energy difference between isomers **III** and **IV** for *k* = 2 is only 7 kJ·mol^−1^. Hence positional {C=C ↔ B(H_2_)B} substitutions might have important changes in the interaction of these hybrid boron-carbon isomers with external perturbations, such as photochemical processes.

We follow with another conjugated hydrocarbon with *n* = 8 π electrons: benzocyclobutadiene and the hybrid boron-carbon series C_(8−2*k*)_B_(2*k*)_H_(6+2*k*)_, with *k* = {0, 1, 2, 3, 4}, similar to pentalene, and displayed in Table 3a–c and Figure 3a–c. Benzocyclobutadiene polymerises readily and reacts as a dienophile in Diels-Alder reactions [37]. As opposed to the previous *n* = 8 systems, in this particular case we need to consider different Kekulé structures, *K*_1_, *K*_2_ and *K*_3_ (Scheme 1) since they give different structural isomers on consecutive *k* {C=C ↔ B(H_2_)B} substitutions. The energy gap in the original hydrocarbon is 195 kJ·mol^−1^ (the same for *K*_1_, *K*_2_ and *K*_3_) and we find for the first time that {C=C ↔ B(H_2_)B} substitutions lead to hybrid systems with lower gaps, e.g., isomer **IV** with *k* = 2 in Table 3a with a gap of 108 kJ·mol^−1^, the lowest among all as shown in Figure 3a, clearly due to the presence of the strained cyclobutadiene ring in the structure. Only isomers from Kekulé structures *K*_1_ and *K*_3_ keep this tendency as shown in Table 3a–c respectively. There is an exception to this behavior: for the Kekulé structure *K*_2_—Table 3b—the gaps are always larger in the substituted isomers, as compared to benzocyclobutadiene. This is striking and due to the presence of only one double bond C=C in the rectangular cyclobutadiene, hence lowering the strain energy on any further {C=C ↔ B(H_2_)B} substitution. Another noteworthy result is the singlet-triplet energy gap differences in the boron equivalent structures of *K*_1_, *K*_2_ and *K*_3_, always fairly above benzocyclobutadiene by 70 kJ·mol^−1^, 115 kJ·mol^−1^ and 180 kJ·mol^−1^ respectively, as shown in Figure 3a–c.

There are many isomers with larger energy gaps as compared to benzocyclobutadiene: The borane structure for planar (*k* = 4) B_8_H_14_ in the *K*_3_ configuration—Table 3c and Figure 3c—has the largest gap of all structures and isomers, 374 kJ·mol^−1^. When strain is released from the rectangular cyclobutadiene moiety in the original hydrocarbon, many isomers have gaps above 300 kJ·mol^−1^, with the interesting case of isomer **I** of *K*_2_ with *k* = 2—Table 4b—a *cis*-butadiene system fused with two diborane molecules and energy gap 368 kJ·mol^−1^ and the lowest ground state energy. The experimental vertical singlet-triplet gap in butadiene is 310.7 kJ·mol^−1^ [38] (318.9 kJ·mol^−1^ with a B3LYP/cc-pVTZ computation). Therefore, the two diborane molecules fused to the *cis*-butadiene moiety in isomer **I** of C_4_B_4_H_10_ (*k* = 2) and Kekulé structure *K*_2_ stabilise the system further by 50 kJ·mol^−1^. We do not intend to give spectroscopic accuracy in these examples, but rather an explanation of the tendency in energy and stability changes upon {C=C ↔ B(H_2_)B} substitutions in conjugated hydrocarbons. We should emphasize here that all structures and isomers remain planar upon these substitutions in benzocyclobutadiene, checked through frequency computations, thus following the previous examples with the exception of cyclooctatetraene, which is not planar.

Given the topological distribution of single and double bonds in benzocyclobutadiene, only Kekulé structures *K*_1_ and *K*_3_ retain the cyclobutadiene ring in the hybrid boron-carbon series. Hence, for *K*_1_—Table 3a and Figure 3a—one has isomer **III** from C_6_B_2_H_8_ (*k* = 1), with an energy gap of 118 kJ·mol^−1^ and isomer **IV** from C_4_B_6_H_10_ (*k* = 2) with a gap of 108 kJ·mol^−1^ (lowest gap). These isomers are equivalent in *K*_3_—Table 3c and Figure 3c—to isomer **II** from C_6_B_2_H_8_ (*k* = 1), and isomer **IV** from C_4_B_6_H_10_ (*k* = 2). The presence of the cyclobutadiene ring explains why these hybrid boron-carbon isomers have such low singlet-triplet energy gaps.

We turn now to the next aromatic system, naphthalene, with *n* = 10 π electrons, and the series C_(10−2*k*)_B_(2*k*)_H_(8+2*k*)_, with *k* = {0, 1, 2, 3, 4}. Naphthalene is a white crystalline solid with a characteristic odor and detectable at very low concentrations and reacts more readily than benzene in electrophilic aromatic substitution reactions [37]. The two Kekulé structures *K*_1_ and *K*_2_ shown in Scheme 1 are also necessary in order to include all possible different isomers upon consecutive {C=C ↔ B(H_2_)B} substitutions. Table 4a,b and Figure 4a,b include the energies and energy gaps for all isomers derived from Kekulé structures *K*_1_ and *K*_2_ in naphthalene, respectively. What differs notably, *viz.*, the energy gaps as compared to antiaromatic systems (*n* = 4, 8)—see above also benzene—is the small difference between the gaps in the original conjugated hydrocarbon C_10_H_8_ and the equivalent borane structure B_10_H_18_, 10 kj·mol^−1^ and 3 kJ·mol^−1^ higher for Kekulé structures *K*_1_ and *K*_2_ respectively. Therefore, from a thermochemical point of view, the synthesis of the planar borane structures derived from benzene and naphthalene, B_6_H_12_ and B_10_H_18_ respectively, should be affordable.

Again, for an aromatic system, consecutive {C=C ↔ B(H_2_)B} substitutions lead to larger energy gaps compared to naphthalene, with a striking increase for the *k* = 2 isomer **I** in structures *K*_1_ and *K*_2_—both are equivalent—C_6_B_4_H_12_ (*k* = 2)—Table 4a—which corresponds to a fusion of two diborane molecules to benzene, and giving a further stability to the molecule. Isomers with lower gaps than naphthalene appear both in *K*_1_ and *K*_2_ structures. The two lowest gaps in *K*_1_ correspond to isomers **II** from C_6_B_4_H_12_ (*k* = 2) and isomer **I** from C_4_B_6_H_14_ (*k* = 3). The lowest gap in *K*_2_—Table 4b—correspond to the *k* = 1 isomer **III** from structure C_8_B_2_H_10_; this destabilisation is clearly due to the loss of aromaticity in the whole system on substituting one C=C bond by a B(H_2_)B moiety on the right ending of the molecule, thus quenching the resonance energy.

We should also emphasize that *K*_1_ set has more isomers than the *K*_2_ set with a gap larger than naphthalene, and all isomers are planar structures corresponding to energy minima for all set of Kekulé structures.

Finally, the last system included in this work, aromatic azulene with *n* = 10 π electrons follows the same series as in naphthalene: C_(10−2*k*)_B_(2*k*)_H_(8+2*k*)_, with *k* = {0, 1, 2, 3, 4, 5}. Two terpenoids of azulene appear in Nature and offer a rich organic chemistry [39]. In azulene we have one, 5 and 10 isomers for *k* = {0, 5}, *k* = {1, 4} and *k* = {2, 3} respectively, as shown in Table 5 and Figure 5. There is only one Kekulé structure leading to different isomers, and the presence of a seven-membered ring fused to a five-membered ring entails a different electronic structure as depicted by the lower energy gap in the original hydrocarbon as compared to benzene and naphthalene.

The azulene equivalent borane structure B_10_H_18_ has a singlet-triplet energy gap 100 kJ·mol^−1^ above the original system and again there are isomers with lower and higher energy gaps as compared to azulene. The largest and lowest gaps correspond to 335 kJ·mol^−1^ and 163 kJ·mol^−1^ for isomer **III** of structure C_4_B_6_H_14_ (*k* = 3) and isomer **X** of structure C_6_B_4_H_12_ (*k* = 2), respectively, as displayed in Figure 5, with the encircled numbers. The general rule is that the average gap increases with *k* for 1 < *k* < 4, namely, the more {C=C ↔ B(H_2_)B} substitutions, the more stable the system on average. One {C=C ↔ B(H_2_)B} substitution in azulene, leading to five C_8_B_2_H_10_ isomers (*k* = 1), shows two isomers (**I**, **III**) with an energy gap above azulene and three isomers (**II**, **IV** and **V**) with energy gaps below azulene. The energy gap in isomer **III** is only 4 kJ·mol^−1^ higher than in azulene. Note that the structural difference between isomer **I** and isomer **V** of structure C_8_B_2_H_10_ (*k* = 1) stems from the boron substitution site: pentagon and heptagon respectively, and both on the bridge atomic site. Boron substitution on the pentagon cycle at the bridge position leads to the least reactive species (*k* = 1) C_8_B_2_H_10_ (**I**) as compared to azulene. However, this situation is inverted when the boron substitution site is on the bridge position of the heptagon cycle, leading to the more reactive isomer **V** for C_8_B_2_H_10_ (*k* = 1) as compared to azulene. As commented above, complete boron substitution of azulene (*k* = 4) leads to a more stable structure, with a singlet-triplet gap 100 kJ·mol^−1^ above azulene. This is remarkable if we take into account that B_10_H_18_ has never been synthesized.

## 3. Discussion

The experimental CC and BB distances in ethylene and diborane(6) are 1.340 Å and 1.736 Å respectively. These are considerable differences which are due mainly to the presence of two bridge hydrogen atoms in diborane(6), changing the electronic structure of the substituted systems considerably. However, the role of these two bridge hydrogen atoms in the B(H_2_)B moiety correspond, from a structural and electronic point of view, to the two π electrons in ethylene: Every {C=C ↔ B(H_2_)B} substitution in the original conjugated C*_n_*H*_m_* hydrocarbon leads to a hybrid planar or nonplanar equivalent structure, as we have seen above in the description of the results. The aromaticity and antiaromaticity of a conjugated C*_n_*H*_m_* system according to Hückel’s 4*n*+2 and 4*n* rule for π electrons, respectively, does not necessarily apply by consecutive {C=C ↔ B(H_2_)B} substitutions, leading to the hybrid systems C_(*n*−2*k*)_B_(2*k*)_H_(*m*+2*k*)_, with *k* = {0, 1, 2, …, *n*/2}. The geometrical parameters for cyclobutadiene, benzene and cyclooctatetraene and the hybrid boron-carbon systems are gathered in Table 6.

According to Table 6, inclusion of empirical dispersion corrections in the B97D functional leads to slightly larger distances in a systematic way as compared to the B3LYP results, without changing the minimum energy nature of the stationary points for the conjugated hydrocarbon structures and the corresponding boron-carbon hybrids.

As regards to the singlet-triplet energy gaps in the same three model systems, we gathered in Table 1 the B3LYP and B97D energy gaps, with a systematic lowering when including dispersion corrections, and following the same trend; since our goal here is a tentative prediction, from a theoretical point of view, of the stabilities in hybrid boron-carbon systems which are experimentally unknown, the qualitative description is valid, without pretending spectroscopic accuracy.

A further assessment on the degree of aromaticity in the hybrid boron-carbon systems included in this work can be tackled with the comparison of the nucleus-independent chemical shifts (NICS) [40], at the centre of ring—NICS(0)—and 1 Å above this point, perpendicular to the ring—NICS(1). NICS is a computational method which provides the extent of absolute magnetic shielding at the centre of a ring. The values are reported with a reversed sign to make them compatible with the chemical shift conventions of NMR spectroscopy, and negative NICS values indicate aromaticity and positive values antiaromaticity. In Table 7 we gather the NICS(0) and NICS(1) values for cyclobutadiene and benzene with the corresponding hybrids.

Cyclobutadiene is the most antiaromatic system, and the NICS values decrease with more {C=C ↔ B(H_2_)B} substitutions. Clearly, benzene is the most aromatic system considered here and the first hybrid with one {C=C ↔ B(H_2_)B} substitution retains certain degree of aromaticity. A further {C=C ↔ B(H_2_)B} substitution leads to a positive NICS(0) and a small negative NICS(1). Complete {C=C ↔ B(H_2_)B} substitutions in benzene leads to a non-aromatic cyclic hexaborane(12) B_6_H_12_. The ACID plots (Table 1) of the benzene derivatives show that each {C=C ↔ B(H_2_)B} substitution produces gaps in the isosurface as indication of reduced aromaticity. In Figure 6 we display the correlation between the singlet-triplet energy gap vs. NICS(0) and NICS(1) from Table 7. The correlations are quite linear except for the two encircled hybrid structures.

The most noticeable changes upon {C=C ↔ B(H_2_)B} substitutions for a given conjugated hydrocarbon C*_n_*H*_m_*, from an energy point of view and singlet-triplet gap, is in the loss of aromaticity or in the inclusion or permanence of very unstable moieties, such as cyclobutadiene. For instance, for antiaromatic cyclobutadiene and cyclooctatetraene—Table 1 and Figure 1—the minimum and maximum singlet-triplet energy gaps correspond respectively to the conjugated hydrocarbon C*_n_*H*_m_* and the corresponding borane systems B*_n_*H*_m_*_+*n*_ respectively, namely, with all *k* = *n*/2 {C=C ↔ B(H_2_)B} substitutions. This is remarkable, since planarity of borane molecules is an exception rather a rule. The loss of aromaticity in benzene upon one {C=C ↔ B(H_2_)B} substitution leads to the minimum energy gap structure, with the largest gap for benzene itself. Pentalene has the lowest energy gap as shown in Table 2 and Figure 2, as an antiaromatic system, with the larger gap for the *k* = 3 isomer (I), where only one C=C double bond remains in the structure, releasing the constrained hydrocarbon structure due to longer BB and B(H_b_)_2_B distances. The particular case of benzocyclobutadiene, also antiaromatic, is remarkable due to the three different Kekulé structures leading to different isomers, Table 3a–c and Figure 3a–c. Thus, for *K*_1_, the minimum singlet-triplet gap comes from the *k* = 2, isomer (IV), which is nothing but a cyclobutadiene molecule with two diborane(6) moieties attached to one of the C=C double bonds forming a six-member cycle: This isomer has a gap which is even lower than in cyclobutadiene itself! On the other hand, and following the cases of cyclobutadiene and cyclooctatetraene, the complete borane structure (*k* = 4) B_6_H_14_ has the largest gap. For Kekulé structure K_2_, benzocyclobutadiene itself has the lowest gap and the largest gap corresponds to *k* = 2 isomer (I), a *cis*-butadiene molecule with two diborane(6) molecules fused and forming a six- and four-member ring on the =CH_2_ and =CH- moieties respectively. Hence the conjugation of the two double bonds is maintained even with the inclusion of diborane moieties. As for *K*_3_, the minimum gap corresponds to the *k* = 2 isomer (IV) which is cyclobutadiene with two fused diborane molecules forming a six-member ring; note that whenever a cyclobutadiene is maintained in the {C=C ↔ B(H_2_)B} substitutions the energy gap is very, thus predicting a very reactive system. Again, the maximum energy gap corresponds the completed borane structure with four {C=C ↔ B(H_2_)B} substitutions: This is a rule for antiaromatic systems.

We now turn to aromatic naphthalene with the *K*_1_ and *K*_2_ Kekulé structures as displayed in Table 4a–b and Figure 4a,b. Due to the indistinguishable nature of the two valence-bond Kekulé structures in benzene, the *k* = 2 isomer (I) is equivalent for *K*_1_ and *K*_2_ and corresponds to a benzene molecule with two diborane molecules fused forming an additional six-member ring. If we use a valence-bond or multiconfigurational wave function, these isomers would have a different energy due to the different spin-coupling patterns. In our approximation, this isomer has the largest energy gap due to the aromaticity of benzene, which is only 14 kJ·mol^−1^ higher than in benzene. The lowest energy gaps correspond to structures where the cyclic aromaticity is somehow broken due to {C=C ↔ B(H_2_)B} substitutions in concrete places: for K_1_ this corresponds to *k* = 2 isomer (II), with three consecutive C=C bonds passing through the bridge C=C bond but with two B(H_2_)B moieties on opposite rings, thus destroying the cyclic aromaticity. As for *K*_2_, the lowest gap corresponds to the *k* = 1 isomer (III) and similarly the aromaticity is destroyed by one {C=C ↔ B(H_2_)B} substitution on the ending edge. We should emphasize that in this latter case the energy gap is much lower as compared to the minimum gap structure for *K*_1_ since the three consecutive alternating C=C bonds gives further stability of the *k* = 2 isomer (II) structure.

Finally, azulene boron-carbon hybrids give a range of singlet-triplet gaps from 163 kJ·mol_-1_ to 335 kJ.mol^−1^, corresponding to structures *k* = 2 isomer (X) and k = 3 isomer (III) respectively, as shown in Table 5 and Figure 5. Due to the larger seven-member ring the structures are less constrained, but the presence of a cyclopentadiene ring lowers the stability. The lowest energy gap structure—*k* = 2 isomer (X)—consists of a cyclopentadiene ring where one of the connecting atoms is a boron atom; this structure should be very reactive indeed.

## 4. Computational Methods

The quantum-chemical computations of the structures and isomers included in this work were carried out with the B3LYP/cc-pVTZ model [41,42,43,44] and with the scientific software Gaussian16 [45]. All geometries correspond to energy minima, checked through frequency computations, with geometry optimisation thresholds of 0.00045 Hartree/Bohr and 0.00030 Hartree/Bohr for maximum force and root-mean-square (RMS) force respectively, and 0.0018 Bohr and 0.0012 Bohr for maximum displacement and RMS displacement respectively. For the computations of the D_2d_ → D_4h_ barriers in cyclooctatetraene C_8_H_8_ and the boron substituted system B_8_H_16_, the optimized geometries showed zero and one imaginary frequency for the D_2d_ and D_4h_ geometries, respectively. The MEP and ACID plots for benzene and the C_(6-2*k*)_B_(2*k*)_H_(6+2*k*)_ series, *k* = {1–3} (Table 1), were also computed with Gaussian16.

## 5. Conclusions

The goal of this work was to check the structural stability from any conjugated hydrocarbon C*_n_*H*_m_* through the boron-carbon hybrid series C_(*n*-2*k*)_B_(2*k*)_H_(*m*+2*k*)_, obtained upon *k* {C=C ↔ B(H_2_)B} substitutions for *k* = {0, 1, 2, …, *n*/2}, leading to the structurally equivalent complete borane B*_n_*H*_m_*_+*n*_ structure, for *k* = *n*/2. We have chosen planar and non-planar conjugated hydrocarbons: cyclobutadiene (*n* = 4), benzene (*n* = 6), cyclooctatetraene (*n* = 8), pentalene (*n* = 8), naphthalene (*n* = 10) and azulene (*n* = 10). As a result, all hybrid boron-carbon structures appear as energy minima from the quantum-chemical computations and therefore, from a thermodynamic point of view, they should be a synthetic target for experimentalists working on planar boron chemistry. This is unusual if we consider that borane polyhedral molecules are 3D curved structures. In the particular case of cyclooctatetraene C_8_H_8_, a non-planar structure, we have shown that the one-to-one structural equivalence C*_n_*H*_m_* ↔ B*_n_*H*_m_*_+*n*_ also holds; the energy barrier for loss of planarity from a planar transition state structure (*D_4h_*) to an energy minimum twisted structure (*D_2d_*) is larger for the cyclic borane B_8_H_16_, as compared to cyclic octatetraene. Structural isomers appear for a given number *k* of {C=C ↔ B(H_2_)B} substitutions in an isoelectronic series C_(*n*-2*k*)_B_(2*k*)_H_(*m*+2*k*)_ with *k* = {0, 1, 2, …, *n*/2} for all conjugated hydrocarbons, except for cyclobutadiene and benzene, with a maximum number of isomers of 10 for azulene structures with C_6_B_4_H_12_ (*k* = 2) and C_4_B_6_H_14_ (*k* = 3) formulae. These isomers are ordered in increasing energy {**I**, **II**, …, *N_iso_*}. Striking energy differences in an isomeric series stem from one {C=C ↔ B(H_2_)B} substitution in the cyclobutadiene cycle from benzocyclobutadiene with an energy jump of 125 kJ·mol^−1^. As regards to the energy order of vertical singlet-triplet energy gaps, they do not necessarily follow the same trend as compared to the energy profile for the isomers of a given *k* in a C_(*n*-2*k*)_B_(2*k*)_H_(*m*+2*k*)_ structure.

The vertical singlet-triplet energy gaps in conjugated hydrocarbons C*_n_*H*_m_* vs. B*_n_*H*_m+n_* change strikingly in antiaromatic systems, when the number of π electrons is 4*n*, with the energy gap always much larger for the borane systems. On the other hand, for conjugated hydrocarbons with (4*n* + 2) electrons—Hückel rule for aromaticity—the difference in vertical singlet-triplet energy gaps between C*_n_*H*_m_* and B*_n_*H*_m+n_* is minor for benzene and naphthalene and larger for azulene. From these results a potential rule arises, which will be thoroughly checked in a future work and holds for the systems included here: the lowest and largest singlet-triplet energy gap for a C_(*n*-2*k*)_B_(2*k*)_H_(*m*+2*k*)_ series of structures—with *n* multiple of 4—always corresponds to the *k* = 0 and *k* = *n*/2 structure respectively, namely, to the original conjugated hydrocarbon C*_n_*H*_m_* and the complete borane structure B*_n_*H*_m_*_+*n*._ We should also emphasize that, if aromatic structures—e.g., π sextets in six-member rings—are maintained in a given isoelectronic C_(*n*-2*k*)_B_(2*k*)_H_(*m*+2*k*)_ series, then the singlet-triplet energy gap is large and similar to the original conjugated hydrocarbon and the structure is predicted to be stable. On the other hand, the presence of an antiaromatic structure—e.g., cyclobutadiene moieties with 4π electrons—in any C_(*n*-2*k*)_B_(2*k*)_H_(*m*+2*k*)_ system, lowers considerably the energy gap thus predicting an unstable structure. Finally, the existence of similar structures derived from phenanthrene derivatives with one {C=C ↔ B(H_2_)B} substitution in the conjugated hydrocarbon [19,20,21,22,23], gives support to the possibility of synthesizing or characterizing the hybrid boron-carbon isoelectronic C_(*n*-2*k*)_B_(2*k*)_H_(*m*+2*k*)_ structures presented in this work.

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
