# Peer review of "Hybrid Boron-Carbon Chemistry"

_molecules, 2020, doi:10.3390/molecules25215026_

Round 1

Reviewer 1 Report

The paper describes computational work to assess the stability of aromatic and antiaromatic hydrocarbon ring systems where C=C units are sequentially replaced by B(H2)B moeties.

While the language and computation looks sound, the major problem I have with this manuscript is that it simply describes the computational results (namely in terms of singlet-triplet energy gaps for a series of aromatic (or antiaromatic) hydrocarbons, but never really discusses the ideas or reasons behind the observed trends; in essence this paper is missing a whole section on discussion, which should seek to explain these computational results in more detail. For example, the results for the k = 3 structure based on benzene (B6H12) are disclosed, but it would have been helpful to place them in the context of the observed B6H12 structure based on Wade's Rules. The same could be said for many of the other structures analyzed.

On a minor note, I think the plot shown in Figure 1 is rather pointless as a correlation can not be reported for just two data points.

Author Response

We thank the Reviewer for his/her comments.

  • The paper describes computational work to assess the stability of aromatic and antiaromatic hydrocarbon ring systems where C=C units are sequentially replaced by B(H2)B moieties. While the language and computation looks sound, the major problem I have with this manuscript is that it simply describes the computational results (namely in terms of singlet-triplet energy gaps for a series of aromatic (or antiaromatic) hydrocarbons, but never really discusses the ideas or reasons behind the observed trends; in essence this paper is missing a whole section on discussion, which should seek to explain these computational results in more detail.

Answer: We have included a comprehensive Discussion section, from page 21 to page 24 in the revised manuscript, where geometries, energies and singlet-triplet gaps are discussed from the obtained results in the previous section.

  • For example, the results for the k = 3 structure based on benzene (B6H12) are disclosed, but it would have been helpful to place them in the context of the observed B6H12 structure based on Wade's Rules. The same could be said for many of the other structures analysed.

Answer: Up to our knowledge, the B6H12 structure – planar cyclic hexaborane(6) – has not been observed. As far as we understand Wade’s rules apply for polyhedral (car)boranes with a 3D curved structure, which is not the case here, since all systems are planar, except for the cyclooctatetraene hybrid boron-carbon series. In his seminal paper, Wade, K. (1971). "The structural significance of the number of skeletal bonding electron-pairs in carboranes, the higher boranes and borane anions, and various transition-metal carbonyl cluster compounds". J. Chem. Soc. D. 1971: 792–793, planar borane systems or systems corresponding to concatenation of B(H2)B moieties in a (poly)cyclic fashion were not considered.

  • On a minor note, I think the plot shown in Figure 1 is rather pointless, as a correlation cannot be reported for just two data points.

Answer: We have deleted this plot.

Reviewer 2 Report

Comments to the Authors of the manuscript molecules-954542

The manuscript entitled: “Hybrid Boron-Carbon Chemistry” reports results of DFT calculations of molecule energy, structure and singlet-triplet energy gap of conjugated hydrocarbons and their boron analogs . The aim of the study was to compare the stability of conjugated hydrocarbon CnHm with the boron hybrid systems C(n−2k)B(2k)H(m+2k). The authors chose ten model molecules (cyclic conjugated hydrocarbons) and modified them by replacing C=C  by B(H2)B moieties.

The undertaken research topic is interesting, and its aim is to obtain new materials for non-linear optics or nanotechnology applications.

However, in the section entitled “Results and discussion”, there is no discussion of the results obtained. Only the energies of the optimized structures and the singlet-triplet gaps are given. On this basis, it is impossible to determine the properties of the tested molecules. The authors should at least calculate the energy difference of the HOMO and LUMO orbitals. The aromaticity of the test molecules is also important. To describe it, for example, the magnetic parameter NICS should be determined.

In “Conclusions”, no conclusions from the performed calculations were given.

Reviewer 3 Report

The authors present a thorough computational study of conjugated hydrocarbons and their isoelectronic substituted boronic compounds. The main quantities analysed are the isomers' stabilities and the singlet-triplet gaps, identifying some trends upon substitution which could be of potential interest in the design of new materials or in rationalising the reactivity of such B containing rings.

The presentation is adequate with a well-structured discussion. I generally recommend publication, after considering the points listed below:

major points:

  • the DFT calculations were carried out without dispersion corrections. This has literally no effect in quantities such as the vertical singlet-triplet gap, but could have a significant impact on the isomers' relative energies (one should remember the failure of DFT in the description of isodesmic reactions). If the calculations are to not be carried out with some simple correction such as Grimme's D3, then some sample values should be provided showing that the effect is minimal.

minor comments:

  • "groundstate" should read as "ground state"
  • the basis set chosen is rather wasteful. The Dunning basis sets were developed for correlated wave function calculations, not with DFT in mind. The same level of accuracy could have probably been achieved with a Karlsruhe def2-TZVP basis at a lower computational cost. Just a comment, nothing needs to be changed.
  • it would be of interest to signal the structures which have been synthetised. My suggestion would be to include some sort of graphical element in the compounds' figures.
  • the authors should be explicit about the optimisation thresholds used in the structure optimisations.

Author Response

Reply to Reviewer 3

We are grateful to the Reviewer for his/her comments.

  • The DFT calculations were carried out without dispersion corrections. This has literally no effect in quantities such as the vertical singlet-triplet gap, but could have a significant impact on the isomers' relative energies (one should remember the failure of DFT in the description of isodesmic reactions). If the calculations are to not be carried out with some simple correction such as Grimme's D3, then some sample values should be provided showing that the effect is minimal.

Answer: We have carried out B97D/cc-pVTZ computations which include Grimme’s dispersion corrections, for the cyclobutadiene, benzene and cyclooctatetraene boron-carbon hybrid series which are included in revised Table 1, page 4 of the revised manuscript. The B97D functional gives systematically lower singlet-triplet energy gaps as compared to the B3LYP functional, but following the same trends and the optimised structures are the same, with slightly increased bond distances for the B97D results as compared to B3LYP, as included in new Table 6 in the added Discussions section. Elongation of distances – though minor – are in agreement with lower singlet-triplet gaps. The gap decreases considerably for the BnHm+n structures with the B97D functional (47 kJ·mol-1 for cyclic B6H12) and this behaviour, probably due to the H-bridge atoms in the B(H2)B moieties, will be considered in a future work.

 minor comments:

  • "groundstate" should read as "ground state"

Answer: corrected.

  • The basis set chosen is rather wasteful. The Dunning basis sets were developed for correlated wave function calculations, not with DFT in mind. The same level of accuracy could have probably been achieved with a Karlsruhe def2-TZVP basis at a lower computational cost. Just a comment, nothing needs to be changed.

Answer: We thank the reviewer for this comment and will consider using the mentioned basis set in the future.

  • It would be of interest to signal the structures which have been synthetised. My suggestion would be to include some sort of graphical element in the compounds' figures.

Answer: Only structures derived from phenanthrene – not included in the work – have been characterised experimentally. This is emphasized in the Introduction, bottom of page 1, refs [19-23], and in the conclusions, page 25, point 10.

  • The authors should be explicit about the optimisation thresholds used in the structure optimisations.

Answer: We have included the optimisation thresholds in the Computational Methods, section 4, page 24.

“…All geometries correspond to energy minima, checked through frequency computations, with geometry optimisation thresholds of 0.00045 Hartree/Bohr and 0.00030 Hartree/Bohr for maximum force and root-mean-square (RMS) force respectively, and 0.0018 Bohr and 0.0012 Bohr for maximum displacement and RMS displacement respectively.”              

Reviewer 4 Report

The authors presented systematic studies on B(H2)B substituted polycyclic conjugated hydrocarbons. Especially, singlet-triplet gaps were computed for the stabilities/reactivities for the species. 

This paper can be improved if the authors consider the following two points.

1) Effects of dispersion corrections. The authors can use B3LYP+D3(BJ) functionals and describe whether the conclusions change or not.

2) The many physical organic chemists used NICS for the aromaticity. This reviewer will be grateful if the authors made the comments of the relationship of the NICS method with this S-T gaps in some cases.

In addition, the authors used a term (relative energy) ΔE(1, N) on the captions of Figures 1-5, and no definition of N. Does that mean ΔE(I,N) because they used Roman numerals for the numbers of isomers. However, the Roman numerals are for many k's (k = 1,2,3,4..). Hence the definition is not clear.

Cartesian coordinates of optimized geometries must be shown in the Supplementary Material.

On the graphical TOC, "2" of "B(H2)B" must be subscripted.

Author Response

We are grateful to the Reviewer for his/her comments.

Round 2

Reviewer 2 Report

The current version of the manuscript is much better than the original. The authors took into account all my comments, therefore I have no objections.

Author Response

Please find attached the reply to Reviewer 2

Reviewer 3 Report

I thank the authors for the reply and would accept the manuscript in its current form.

Author Response

We are grateful to the Reviewer for his/her comments.

  • The DFT calculations were carried out without dispersion corrections. This has literally no effect in quantities such as the vertical singlet-triplet gap, but could have a significant impact on the isomers' relative energies (one should remember the failure of DFT in the description of isodesmic reactions). If the calculations are to not be carried out with some simple correction such as Grimme's D3, then some sample values should be provided showing that the effect is minimal.

Answer: We have carried out B97D/cc-pVTZ computations which include Grimme’s dispersion corrections, for the cyclobutadiene, benzene and cyclooctatetraene boron-carbon hybrid series which are included in revised Table 1, page 4 of the revised manuscript. The B97D functional gives systematically lower singlet-triplet energy gaps as compared to the B3LYP functional, but following the same trends and the optimised structures are the same, with slightly increased bond distances for the B97D results as compared to B3LYP, as included in new Table 6 in the added Discussions section. Elongation of distances – though minor – are in agreement with lower singlet-triplet gaps. The gap decreases considerably for the BnHm+n structures with the B97D functional (47 kJ·mol-1 for cyclic B6H12) and this behaviour, probably due to the H-bridge atoms in the B(H2)B moieties, will be considered in a future work.

 minor comments:

  • "groundstate" should read as "ground state"

Answer: corrected.

  • The basis set chosen is rather wasteful. The Dunning basis sets were developed for correlated wave function calculations, not with DFT in mind. The same level of accuracy could have probably been achieved with a Karlsruhe def2-TZVP basis at a lower computational cost. Just a comment, nothing needs to be changed.

Answer: We thank the reviewer for this comment and will consider using the mentioned basis set in the future.

  • It would be of interest to signal the structures which have been synthetised. My suggestion would be to include some sort of graphical element in the compounds' figures.

Answer: Only structures derived from phenanthrene – not included in the work – have been characterised experimentally. This is emphasized in the Introduction, bottom of page 1, refs [19-23], and in the conclusions, page 25, point 10.

  • The authors should be explicit about the optimisation thresholds used in the structure optimisations.

Answer: We have included the optimisation thresholds in the Computational Methods, section 4, page 24.

“…All geometries correspond to energy minima, checked through frequency computations, with geometry optimisation thresholds of 0.00045 Hartree/Bohr and 0.00030 Hartree/Bohr for maximum force and root-mean-square (RMS) force respectively, and 0.0018 Bohr and 0.0012 Bohr for maximum displacement and RMS displacement respectively.”              

Reviewer 4 Report

The authors added new results including NICS and corrected errors. This reviewer recommends publication without any changes. 

Author Response

Please find attached reply to Reviewer 4
